

# 1 Syn-thrusting, near-surface flexural-slipping and stress deflection

# 2 along folded sedimentary layers of the Sant Corneli-Bóixols Anticline

# 3 (Pyrenees, Spain).

Stefano Tavani[1], Pablo Granado[2,3], Pau Arbués[2,3], Amerigo Corradetti[1], Josep A. Muñoz[2,3]
[1] DiSTAR, Università degli Studi di Napoli "Federico II", Largo S. Marcellino 10, 80138 Naples, Italy
[2] Institut de Recerca Geomodels, Universitat de Barcelona, Martí i Franquès s/n, 08028 Barcelona, Spain
[3] Departament de Dinàmica de la Terra i de l'Oceà, Universitat de Barcelona, Martí i Franquès s/n, 08028 Barcelona, Spain
*Correspondence to*: Stefano Tavani (stefano.tavani@unina.it)
**Abstract.** In the Spanish Pyrenees the Sant Corneli-Bóixols thrust-related anticline displays an outstandingly preserved
growth strata sequence. These strata lie on top of a major unconformity exposed at the anticline's forelimb that divides and
decouples a lower pre-folding unit from an upper syn-folding one. The former consists of steeply-dipping to overturned
strata with widespread bedding-parallel shears indicative of folding by flexural-slip, whereas the syn-folding strata above
define a 200-m amplitude S-shaped fold. In the inner and outer sectors of the forelimb, both pre- and syn-folding strata are
near-vertical to overturned and the unconformity angle ranges from 10° to 30º. In the central portion of the forelimb, syn-
folding layers are shallowly-dipping, whereas the angular unconformity is about 90° and the unconformity surface displays
strong S-C shear structures, which provide a top-to-the foreland shear sense. This sheared unconformity is offset by steeply-
dipping faults which are at low angles to the underlying layers of the pre-folding unit. Strong shearing along the
unconformity surface also occurred in the inner sector of the forelimb with S-C structures providing an opposite, top-to-the
hinterland, shear sense. Cross-cutting relationships and shear senses along the pre-folding bedding surfaces and the
unconformity indicate that regardless of its orientation, layering in the pre- and syn-folding sequences of the Sant Corneli-
Bóixols anticline was continuously sheared. This shearing promoted an intense stress deflection, with the maximum
component of the stress tensor remaining at low angles to beds during most of the folding process.

## 23 1 Introduction

Templates used to describe the state of stress of growing regional-scale thrust-related anticlines (e.g. Hancock, 1985; Lisle,
1994; Fischer and Wilkerson, 2000; Belayneh and Cosgrove, 2004; Tavani et al., 2015), typically integrate punctual strain
data (e.g. Engelder and Geiser, 1980; Laubach, 1989; Lacombe, 2012; Balsamo et al., 2016) and indirect information
provided by the large-scale geometry of the structure, such as like curvature or strata thinning/thickening (e.g. Price and



Cosgrove, 1990). Widespread documentation of bedding-parallel slip, along with the broad preservation of layer thickness,
provide key information to model the distribution of stress in actively growing anticlines. These observations indicate
flexural-slip folding in the multilayered portions of reservoir-scale thrust-related folds (e.g. Donath and Parker, 1964;
Ramsay; 1967; Tanner, 1987; Suppe, 1983; Fowler, 1996; Erslev and Mayborn, 1997). The assumption/observation of
flexural-slipping has important consequences on the stress distribution:

- Shearing along numerous bedding surfaces in a wide range of bedding dip, is possible only where the bedding

surfaces have a low friction and a low cohesion.

- On the other hand, the reactivation of these closely spaced low friction surfaces should prevent the layer-parallel

shear to exceed a certain value, which in turn imposes the direction of the maximum component of the stress field

to be at low angle to bedding, i.e. at low angle to the shearing surface (Wiltschko et al., 1985; Ohlmacher and

Aydin, 1997; Tavani et al., 2015).

The process of layer-parallel shearing is, however, discontinuous in time and space, so that it is unclear whether the regional
stress reorients only locally or, rather, the layer-parallel slipping is sufficiently dense, both in time and space, to promote the
reorientation of the stress in wide - actively folding - areas. Field documentations are in agreement with the second
hypothesis, i.e. that shearing along low-friction bedding surfaces promotes deflection of the principal directions of the
remote stress field, with the direction of the maximum stress keeping at low angle to layers, so that the maximum stress in
active fold and thrust belts maybe not always strictly horizontal. In fact, syn-folding layer-parallel shortening structures are
reported in the folded pre-growth strata of many thrust related anticlines (e.g. Tavani et al., 2006; 2012). However, paleo-
stress and/or paleo-strain indicators cannot easily and unequivocally constrain the bedding dip values range at which such
stress deflection mechanism can operate (e.g. Callot et al., 2010; Beaudoin et al., 2012). In fact, almost all published datasets
come uniquely from pre-growth sequences of thrust-related folds. In these cases, deciphering for which bedding dip values a
given set of deformation structures are formed (i.e. fractures or slickenlines along a bedding surface) remains difficult, and
any assumptions made carry along significant uncertainty.

On the other hand, observations made in syn-growth layers of thrust-related anticlines allow to drastically reducing

uncertainties related to the timing of deformation (e.g. Shackleton et al, 2005, 2011). As a matter of fact, the study of growth
strata sequences are by far the most commonly used approach for understanding the kinematics of fault-related folds in
contractional settings (e.g. Suppe et al., 1992; Burbank et al., 1996; Ford et al., 1997; Suppe et al., 1997; Vergés et al., 2002).
In contrast with the abundance of detailed geometrical studies (Suppe, 1983; Medwedeff, 1989; Mitra, 1990; Suppe and
Medwedeff, 1990; Zapata and Allmendinger, 1996; Poblet et al., 1997; Suppe et al., 1997), few contributions dealing with
the dynamics of folding inferred from syn-kinematic layers have been published (e.g. Ford et al., 1997; Nicol and Nathan,
2001; Shackleton et al., 2011; Beaudoin et al., 2015), mostly because of the lack of well-preserved and accessible exposures.
In this work we have focused on the macro and meso-structures developed within a growth strata wedge and a related major
syn-kinematic unconformity exposed at the forelimb of the Sant Corneli-Bóixols anticline. Bedding-parallel shear occurs
along pre and syn-kinematic strata, which are oriented obliquely to each other, together with meso-scale faults cutting across





strata and the unconformity. Thus, this area provides an excellent, almost unique, field example to observe, describe, and
analyse how anisotropies oblique to each other, i.e. layers and unconformity, respond to progressive shortening and related
folding in a contractional setting. In addition, the studied area allowed us to determine the threshold dip value at which
flexural-slip is of sufficient magnitude to deflect the maximum principal stress direction from the regional stress field.
**2 Geological Setting**
The Pyrenean Belt is a doubly-vergent orogenic wedge (Fig. 1a) formed during the Late Cretaceous to Miocene
subduction of the Iberian lithosphere beneath the Eurasian plate (e.g. Choukroune et al. 1990; Muñoz, 1992; Teixell, 1998).
It largely deformed and inverted the Mesozoic extensional basins developed between Iberia and Eurasia during the Mesozoic
separation of these two plates (Muñoz, 2002). The Early Cretaceous Organyà basin is one of these basins interposed between
the Iberian plate and the exhumed mantle of the Pyrenean rift (e.g. Tugend et al., 2014). Upon convergence and shortening,
the Organyà basin was positively inverted and incorporated into the hanging-wall of the Bóixols thrust starting on Late
Cretaceous times (e.g. Mencos et al., 2015 and references therein). The positive inversion of the inherited extensional
structures occurred under oblique, NNW-SSE oriented, convergence (Tavani et al., 2011), and was responsible for the
development of the E-W striking Sant Corneli-Bóixols anticline (Fig. 1a). The location and geometry of this anticline is
controlled by the orientation of the Early Cretaceous extensional border fault system of the Organyà basin (e.g. Bond and
McClay, 1995; García-Senz, 2002; Mencos et al., 2015).
Several detailed studies of the stratigraphy of the Organyà basin were carried out in the last 50 years (e.g. Rosell,
1963; Garrido; 1973; Simó, 1986; Berástegui et al., 1990; García-Senz; 2002, Mencos, 2011). The pre-rift Mesozoic
stratigraphy is represented by clays and evaporites belonging to the Triassic Keuper facies, followed by Jurassic shallow
marine carbonates and deeper water marls. The Early Cretaceous syn-rift megasequence consists of platform carbonates that
thicken towards the north and transition laterally (i.e. toward the north) into basinal marls (e.g. García-Senz, 2002). In the
hanging-wall of the Bóixols thrust the maximum thickness of the syn-rift megasequence is about 4500m. It thins southwards
around the hinge zone of the Sant Corneli-Bóixols anticline across the extensional fault system at the southern margin of the
Organyà basin (Lanaja et al., 1987; Berástegui et al., 1990; Arbués et al., 1996; García-Senz, 2002; Muñoz et al., 2010;
Mencos et al., 2015). The Upper Cenomanian to Lower Santonian post-rift megasequence consists of carbonates with lesser
clastics and can be up to 700m thick (García-Senz, 2002; Mencos, 2011). The syn-orogenic strata are exposed in the leading
syncline (i.e. Tremp-Sallent Syncline) of the Sant Corneli-Bóixols anticline (Fig. 1a-b) and include more than 1000m of
Upper Santonian to Paleocene deepwater to continental strata that thin abruptly to a few tens of meters northwards, i.e.
towards the Sant Corneli-Bóixols anticline (e.g. Arbués et al., 1996; Roma et al. 2011).
In the studied area, the syn-orogenic succession can be subdivided into two units (Figs. 1b-c): (1) The lower, Upper
Santonian to Campanian, Vallcarga Group (Nagtegaal, 1972) is constituted by a multilayered marine sequence of thin to
medium bedded limestones and mudstones. Around the Sant Corneli-Bóixols Anticline the Vallcarga Group was deposited
during folding as evidenced by growth geometries at the eastern tip of the Sant Corneli Anticline (Mencos et al., 2015).



However, no clear evidences of deposition during the early stages of folding are visible in the studied area and hence it is
here geometrically considered as pre-kinematic but within a regional syn-orogenic scenario. (2) The Late Campanian to
Maastrichtian Areny Group (Arbués et al., 1996) is unconformably overlying the Vallcarga Group. Its thickness exceeds
1000m in the Tremp-Sallent Syncline depocentre (Fig.1a-b) but thins abruptly to a few tens of meters towards the Sant
Corneli-Bóixols Anticline so it is considered to have been deposited during folding (i.e. syn-folding). The Areny Group
records sedimentation in neritic to deep marine conditions coeval with the inversion and related folding of the Organyà
extensional basin. The Areny Group has been divided into four sequences (A1 to A4 from older to younger; Arbués et al.,
1996) and broadly includes rudist accumulations and talus marls, sandstones, and re-sedimented equivalent in turbiditic
facies. (3) The syn-folding Maastrichtian to Paleocene Tremp Group (Cuevas, 1992) includes continental facies associations
which are commonly referred to as the Garumnian facies (Cuevas, 1992). These consist of alluvial and colluvial
conglomerates and breccias, passing southwards to alluvial plain and fluvial reddish sandstones and mudstones (e.g. Arbués
et al., 1996; Roma et al., 2011).
From a structural point of view, the major structures in the studied area are the Bóixols thrust and related splays, the
E-W trending Sant Corneli-Bóixols Anticline and the associated Santa Fe Syncline to the north and the Tremp-Sallent
Syncline to the south (Fig. 1a). The main ramp of the Bóixols thrust crops out in the studied area, whereas to the west and to
the east it remains blind along most of the frontal limb of the Sant Corneli-Bóixols Anticline. In its exposed sector, the
Bóixols thrust has Triassic to Upper Cretaceous pre-growth rocks in its hanging-wall and syn-orogenic and syn-folding strata
in its footwall (Fig. 1c). A thin sheet of overturned post-rift limestones, mainly the upper Cenomanian ones, defines two
thrusts. The lower thrust remains blind beneath the vertical beds of the Garummnian succession at the northern limb of the
Sallent Syncline. The upper one on the other hand truncates the Garumnian beds. This upper one is the Bóixols thrust and
according to magnetostratigraphic and thermochronological studies it would have been reactivated during Paleogene times
(Beamud et al., 2011). The pre-folding beds of the Vallcarga Group in the footwall of the Bóixols thrust are folded into a
syncline with overturned strata immediately below the thrust fault. These strata progressively acquire sub-horizontal attitudes
in the Tremp-Sallent Syncline to the south (Fig. 1c). Conversely, the unconformably overlying Areny and Tremp Groups
display a series of folded structures, namely the Sant Maximí Syncline and the Tremp-Sallent Syncline, with the Remolina
Anticline in between. The Remolina Anticline disappears toward the east, where the two synclines join (Fig. 1b). The three
folds display a significant eastward plunge (Roma et al., 2011) of 24° with a 72° strike, as derived by the direction normal to
the best fit plane of bedding data of the Areny Group (Fig. 1c).

**3 Macro- and Meso-structures**

In the following, we describe the structural assemblages occurring along and around the major unconformity
dividing the Areny from the Vallcarga Group, i.e., the lower syn-folding strata from the upper syn-folding strata (Fig. 1d).
The macro- and meso-structures are described from north to south in three sub-sections, corresponding to the three limbs of
the S-shaped fold that define the Sant Maximí Syncline, the Remolina Anticline, and the Tremp-Sallent Syncline (Fig. 1). We



will present and discuss stereoplots of bedding attitude, fault orientations and kinematic indicators from faults. In these
stereoplots the plane normal to the structural plunge is also displayed, in order to ease the interpretation of the faults
kinematics. For each stereoplot, we also show the two graphs resulting from the removal of plunge first and then of the
residual bedding dip (Ramsay, 1967).

**3.1 Northern limb**

In the inner (northern) limb of the Sant Maximí Syncline, the marls and limestones of the Vallcarga Group are
overturned, while the unconformably overlying strata of the Areny Group are steeply south-dipping to near-vertical (Fig. 2).
In the northern limb of the syncline, the strata of the Areny Group include siltstones, sandstones, and conglomerates
belonging to the A4 sequence; the A1 to A3 sequences are missing (Fig. 2a-b), either because they were never deposited
there or because they have been eroded before the deposition of the A4 sequence. Overturned bedding surfaces of the
Vallcarga Group display evidence of shearing. Movements along bedding are mostly toward the NW with normal sense of
slip in the present overturned bedding orientation (Fig. 2b). Stereoplots show that slickenlines along bedding surfaces of the
Vallcarga Group are mostly perpendicular to the local fold axis. The unconformity between the Vallcarga and the Areny
Groups has been reactivated as a thrust and displays an intense S-C fabric that affects few meters of the Vallcarga Group
(Fig. 2c). Analogously to other S-C tectonites developed in carbonates and at shallow depth (e.g. Tesei et al., 2013; Vitale et
al., 2014), the S-C structures found in the first 2-3 m of the Vallcarga Group immediately below the unconformity, formed as
consequence of pressure-solution of marly limestones and marls, and thus indicating a ductile to brittle-ductile behaviour of
the Vallcarga Group in correspondence of this major fault. Slip directions provided by C, S and C' structures are top-to-the-
NW and, similarly to the bedding-parallel slip surfaces, the average slip direction lies along the plunge-normal plane (Fig.
2c). The Areny Group conglomerates immediately above the unconformity are not affected by S-C fabric, whereas siltstones
occurring few meters above the unconformity are affected by a strong cleavage (Fig. 2d). This cleavage is at high angle to
bedding, as seen in the field and as evidenced by the fact that poles to cleavage in the stereoplot occur close to the bedding
planes great circles (Fig. 2d). However, cleavage is not strictly bedding-perpendicular. Once bedding dip is restored to the
horizontal, the poles to cleavage still lay along the plunge normal-plane and cleavage becomes SE-dipping. These
relationships are indicative of a minor top-to-NW shear component during cleavage development.

**3.2 Central limb**

Immediately to the south of the axial surface of the Sant Maximí Syncline, strata of the Areny Group are shallow dipping to
sub-horizontal (Fig. 1c-d). Locally, however, strata of the A4 sequence of the Areny Group are steeply north-dipping and the
Sant Maximí Syncline forms a tight structure with a north-dipping to near vertical axial surface. In the hinge zone, the
unconformable strata of the Areny Group are separated from the underlying overturned strata of the Vallcarga Group by a
sub-horizontal shear zone, which corresponds to the sheared syn-folding unconformity (Fig. 3). In addition, the sheared
unconformity is offset by a series of high-angle faults that uplift the southern block (Fig. 3b). Striae along bedding surfaces
of the Vallcarga Group indicate top-to-N movements with normal kinematics (Fig. 3b). Striae within the sub-horizontal shear



zone indicate top-to-S movement, whereas the high angle faults that offset it have slickenlines lying along the plunge-normal
plane and show a top-to-N movement. A few strike-slip slickenlines also occur along high angle faults, and are indicative for
left-lateral movements. In other places, the system of high angle reverse faults uplifting the southern limb of the Sant
Maximí Syncline displays a top-to-NW movement, including some right-lateral kinematic indicators (Fig. 3c). Looking at
the system of high angle faults in natural cross-sections at a larger scale of observation (Fig. 3c), it is evident that these faults
are approximately (i.e. angular difference is less than 10°) parallel to the layers of the Vallcarga Group, and that the amount
of uplift of the southern block is few tens of metres.
The uplifted southern block is well exposed at the Sallent hill (Fig. 4). There, the sub-horizontal rudist-bearing units
of the Areny Group A3 sequence sit unconformable on top of the overturned north-dipping strata of the Vallcarga Group
(Fig. 4a-b). Slickenlines are consistently found along the bedding surfaces of these strata. As shown in the stereoplots of
Figure 4a, most of slickenlines display top-to-N normal kinematics, whereas a few are characterised by strike-slip (both left-
and right-lateral) or reverse kinematics. The slickenlines displaying strike-slip and reverse kinematics postdate the top-to-N
normal ones. Faults oblique to bedding have been also found in the Vallcarga Group in this area (Fig. 4a). These faults are at
low angle with the bedding surface and display normal and, subordinately, reverse kinematics. After bedding dip removal,
both bedding-parallel shear surfaces and bedding-oblique faults, show a top-to-NW shear sense. As mentioned above, strata
of the rudist-bearing A3 sequence are shallow dipping (Fig. 4a-b) and the unconformity between the Areny and the Vallcarga
groups is also sub-horizontal. The unconformity is affected by a pervasive shear fabric (Fig. 4c) with S, C, and C' structures
providing a top-to-SSE shear sense. In addition, the sheared unconformity is cross-cut by SSE-dipping and NNW-verging
reverse faults (Fig. 4b).

**3.3 Southern limb**

Strata of the Areny Group are overturned to mostly steeply south-dipping to the south of the Remolina Anticline,
strata of the Areny Group are overturned to mostly steeply south-dipping (Fig. 5a). These strata are still unconformable on
top of the overturned strata of the Vallcarga Group, but the unconformity angle between the two groups becomes
significantly reduced down to about 20°. The unconformity preserves its stratigraphic origin and, as opposed to the northern
and central limbs of the Sant Maximí syncline, no appreciable evidence of shear occurs (Fig. 5b). Instead, striae along the
bedding surfaces of the Areny Group are observed (Fig. 5c). These striae indicate normal top-to-NNW and reverse top-to-
NNE movements along north-dipping overturned and steeply south-dipping strata, respectively (Fig. 5c). In both cases, striae
lie along the bedding surfaces at the intersection between bedding and the plunge-normal plane (Fig. 5c). Faults at low angle
to bedding have the same behaviour as the bedding: south- and north-dipping faults are reverse and normal, respectively,
with slickenlines lying at the intersection between the fault and the plunge normal plane (Fig. 5c). Once bedding dip is
restored to the horizontal, a top-to-NNW shear sense is provided by both faults and bedding-parallel shear surfaces.

**3.4 Structural summary**

The deformation structures observed along and around the unconformity separating the upper syn-folding strata of



the Areny Group from the underlying multilayered limestones and marls of the Vallcarga Group can be summarised as follows:

In the Vallcarga Group, many of the E-W striking bedding surfaces of near-vertical to overturned strata have been reactivated as shear surfaces. Most of these bed-parallel shear surfaces exhibit dip-slip kinematics, with only a few beds showing strike-slip movements. After removing the plunge of the structure and then restoring the local bedding to the horizontal, most of the slickenlines measured along the bedding surfaces provide a shear sense ranging from top-to-NW to top-to-N, with an average top-to-NNW movement. Faults are roughly E-W to WSW-ENE striking and show very low cut-off angles to bedding. After removing the fold plunge and the bedding dip, these faults provide the same top-to-NNW shear sense as the bedding-parallel slickenlines. Some faults, which are presently steeply dipping to near vertical, have cut-off angles of ranging from 20° to 40° and after removing plunge and bedding dip show normal kinematics. Still, the shear sense provided by them after bedding dip removal is top-to-NNW. This fault pattern and the illustrated kinematics of bedding surfaces is observed all across the studied thrust-related fold profile, i.e. in the northern, central, and southern limbs.

The syn-folding unconformity is characterised by an intense S-C fabric (showing also some C' structures) in the northern limb, and in the sub-horizontal central limb. In both cases, shear direction is roughly NNW-SSE, although some strike-slip movements are occasionally observed. However, the shear sense is opposite in the two limbs, being top-to-NNW in the northern limb (i.e., where the sheared unconformity strikes about E-W and has a near vertical attitude) and top-to-SSE in the central limb (i.e., where the sheared unconformity is offset by the steeply-dipping to near vertical faults surging from the underlying Vallcarga Group). Further to the south, in the southern limb of the Remolina Anticline, the unconformity shows little evidence of deformation.

Strata of the Areny Group exposed at the northern limb are affected by an intense cleavage at high angle to bedding. The cleavage-bedding angle is not exactly 90° however, indicating the occurrence of a top-to-NNW bedding-parallel shear component. Slickenlines are observed along the bedding surfaces of the near vertical to overturned strata of the Areny Group exposed at the southern limb of the Remolina Anticline. In this area, some faults at very low angle to bedding occur. For both faults and bedding surfaces, the shear sense measured after removing the plunge and the bedding is roughly top-to-NNW.

## 4 Chronology of deformation stages

The syn-folding strata of the Areny Group exposed at the forelimb of the Sant Corneli-Bóixols Anticline are unconformably on top of north-dipping overturned strata of the pre-folding Vallcarga Group. The unconformity between both groups is clearly a syn-folding feature. Its unconformity angle varies across the studied area, and in addition, its surface shows unequivocal evidence of strong shearing with an average NNW-SSE-oriented shear direction. Such shear direction is parallel to the slip directions measured along both faults and bedding surfaces of the Vallcarga and Areny groups. The NNW-SSE direction is not perpendicular to the strike of the hosting anticline tough. This structural relationship reaffirms that the E-W striking Sant Corneli-Bóixols Anticline has developed under an oblique convergence setting where the shortening



direction was NNW-SSE (Tavani et al., 2011). In this sense, the observed meso-structures are interpreted as developed
during the growth of the Sant Corneli-Bóixols Anticline, and they cannot be attributed to a subsequent tectonic event. The
fact that these structures occur in syn-folding strata, also rules out a pre-folding origin. In agreement with this, the observed
opposite shear senses along the unconformity surface to the north and to the south of the San Maximí Syncline axial surface
have a syn-folding origin. As illustrated in the next section these opposite senses of shear can be used to unravel the
kinematic evolution of the unconformity, the unconformity angle itself and therefore that of the Sant Corneli-Bóixols
Anticline.

## 5 Modelling the folding of the angular unconformity

Guidelines about flexural-folding of angular unconformable sequences where firstly provided by Alonso (1989).

These include the progressive variation of the unconformity angle during tilting of pre-unconformity layers, synchronously
with progressive shearing along the unconformity surface (Alonso, 1989). Figure 6a illustrates the folding of an
unconformable sequence using a kink-band template with synclinal geometry. The position of six key-points undergoing
folding is illustrated, where the fixed $P_1$ point is the origin of our reference system. The simple kink-band construction is
used to quantify how the unconformity angle and the amount of shear along the unconformity are modified during folding,
where $D_0$ is the initial dip of layers, $U_0$ the initial unconformity angle, $H_0$ the stratigraphic elevation of the unconformity, and
$L_0$ the distance from the origin of an arbitrarily placed pin line (note that the $L_0$ parameter will disappear from the final
equations used here). The X and Y coordinates of the 6 key points and the length of the segments joining them can be
expressed as a function of $L_0$, $H_0$, $U_0$, $D_0$, and $D$, as provided in figure 6b. In particular, the length of the segment joining
points $P_5$ and $P_4$ provides the amount of shear ($\Delta S$, considered positive when top-to-the-hinterland), while points $P_3$ and $P_4$
allow calculating the unconformity angle ($U$).

As folding takes place, the unconformity angle increases and the shear sense along the unconformity is initially top-

to-the-hinterland, i.e. in the same sense as the flexural-slip along the pre-unconformity layers. When the pre-unconformity
strata become overturned, the unconformity angle continues to increase and the flexural-slip in the pre-unconformity layers
continues to be top-to-the-hinterland; instead the shear sense along the unconformity flips and becomes top-to-the foreland.
Close to the leading syncline, i.e. along the $P_2P_3$ segment, the incremental shear sense is top-to-the-hinterland, whereas along
the $P_3P_4$ segment it becomes to top-to-the-foreland, despite the cumulative shear sense may continue to be top-to-the-
hinterland.

The relationship between $U$, $U_0$, and $D$ derived in figure 6b, are graphed in figure 6c. The relationships between $U$

on $D$ (blue lines in the figure) for different $U_0$ values, indicate that:

(1) It is possible to develop overturned pre-folding strata and a nearly sub-horizontal unconformity, as observed in

the central limb of our study area, (i.e. the unconformity angle is roughly equal to the dip of the pre-unconformity

layer) for a wide range of initial unconformity angles (from 30° to 90°).



(2) In order to obtain an unconformity angle of less than 20° where the pre-unconformity strata are near vertical, as
observed in the northern and southern limbs of the study area, the initial unconformity angle cannot exceed 10-15°.
The predicted amount and sense of shear (normalised to $H_0$) at different tilting stages (i.e. for different $D$) and for different
initial unconformity angles (i.e. $U_0$) is plotted in red in figure 6c, where positive and negative values indicate top-to-the-
hinterland and top-to-the-foreland shear, respectively. For small initial unconformity angles, the unconformity angle and the
cumulative top-to-the-hinterland slip along the unconformity increase during folding. This occurs until the dip of pre-growth
strata attains a near-vertical attitude. From this point, further folding would imply overturning of strata and the decrease of
the cumulative top-to-the hinterland slip, which eventually becomes negative (i.e. top-to-the-foreland sense), while the
unconformity angle $U$ exceeds 90° Where the initial unconformity angle is instead close to 90°, progressive folding would
imply a short period of top-to-the-foreland shearing along the unconformity, followed by top-to-the hinterland shearing when
strata become nearly vertical to overturned. At this point it is important to remark that the progressive and the incremental
shear senses do not coincide. The $D$ value at which the cumulative shear passes from top to the hinterland to top-to-the-
foreland largely depends on the initial unconformity angle $U_0$. Conversely, for an initial unconformity angle $U_0$ between 45°
to 90° the incremental shear changes its sign for values of $D$ ranging from 80° to 90°, and almost regardless on $U_0$, value (the
regions where the incremental shear has opposite directions are in white and grey in figure 6c).
The absence of any kinematic indicator of top-to-the-hinterland shear in the central limb of the study area, indicates
that the initial unconformity angle had to be high at that position, and if any, the initial stage of top-to-the-hinterland shearing
was negligible. This can be achieved when the initial unconformity angle is at least 70-75°. This represents a key argument
for unravelling the deformation sequence, as structures postdating the top-to-the foreland shearing have to be interpreted as
developed synchronously with layers' tilting, and in particular, as developed at least after layers have become near vertical.
The top-to-the hinterland shear sense and the small unconformity angle observed in the northern limb, instead, point out for a
small initial unconformity angle.
A cautionary note must be added for these conclusions, as they are based on a purely geometric model, in which
both bed thickness and line length preserves during folding. However, based on field observation reported here and in Tavani
et al (2011), pressure solution cleavage is an extremely localized phenomenon in this anticline, and deformation structures
pointing out for folding-related  bed thickness variations are not occurring. In agreement, and despite the intrinsic
simplification of any geometric model, information provided by the model of figure 6 can be applied to our case study.

## 6. Discussion

### 6.1 Relative timing between shearing and layers' tilting

Strata of the Vallcarga Group exposed at the forelimb of the Sant Corneli-Bóixols Anticline display a rather constant
attitude, however, according to the model described in figure 6, shear senses and angles of the unconformity on top of these
strata, indicate that the pre-unconformity layers (i.e. the Vallcarga Group) were not homoclinally-dipping when the Areny



Group was unconformably deposited on top them. The scheme of Figure 7a illustrates the present day simplified geometry of
the studied structure, together with the balanced (i.e. line-length is preserved; Dahlstrom, 1969; Brandes and Tanner, 2014)
reconstruction at a time immediately after the unconformity development. The reconstructed dip of pre-unconformity layers
in the three limbs, is obtained according to what illustrated in section 5. As previously mentioned, the top-to-the-foreland
shearing along the unconformity of the central limb has to be interpreted as occurring when strata of the Vallcarga Group
attained a near-vertical to overturned attitude. On the other hand, the absence of any evidence of top-to-the-hinterland
shearing along the unconformity surface at the central limb, points out that such an unconformity had to be developed when
the layers of the Vallcarga Group were steeply dipping. In agreement with this, the north-dipping faults offsetting the sheared
unconformity have to be regarded as syn-folding structures developed when strata of the Vallcarga Group were overturned.
These 10- to 30-m spaced faults mostly consist of bedding-parallel segments (Figs. 4A and 5d), with some strands showing
cut-off angles between 20° and 40° (see stereoplots of figures 3b-c), and are interpreted as flexural-slip faults, like those
offsetting the topographic surface of growth folds (e.g. Burbank and Anderson, 2011; Gutiérrez et al., 2014; Li et al., 2015).
These faults are therefore late-stage flexural-slip features and, as detailed in the next sub-section, cannot be compatible with
a sub-horizontal maximum stress.

### 6.2 Maximum stress orientation

The studied natural stratigraphic units includes cm to m-thick strata of limestones, marls, sandstones and
conglomerates exposed across an about 500-m-wide area (Fig. 1). The large number of strata involved in the deformation,
coupled with their high compositional variability, prevents the collection of a representative dataset of friction and cohesion
data of both layers and interlayers. It thus makes it impossible to carry out a quantitative dynamic (i.e. stress) reconstruction.
However, many stress configurations can be easily discarded, due to their kinematic inconsistency with the observed
shearing pattern. In particular, we consider that the maximum paleo-stress lies on the plane oriented perpendicular to the
fault/flexural-slip plane and containing the slip direction, and forms an obtuse angle with the shear sense (e.g. Etchecopar et
al., 1981). The following observations can thus restrict the range of possible solution and the sources of stress during folding:
(1) If we consider faults and strata in their present orientation and after plunge and bedding dip removal, the top-to-the-
foreland layer-parallel shearing and the south-verging reverse faulting are rare features in the northern limb of the San
Maximí Syncline, which is located at a distance of less than 100 m from the Bóixols thrust. The scarcity of these structures,
and the occurrence of flexural-slip surfaces forming a low angle with the thrust and having an opposite shear sense (i.e.
normal kinematics), indicates the limited role of faulting-related stress, sourced from the process zone (e.g. Cowie and
Scholz, 1992) of the upward propagating Bóixols thrust, in controlling the pattern of syn-folding shearing. This is contrary to
what has instead been documented in other thrust-related anticlines (e.g. Bellahsen et al., 2006). (2) The top-to-the-hinterland
(i.e. top-to-the-crest of the Sant Corneli-Bóixols Anticline) layer-parallel shearing observed along the bedding surfaces of the
Vallcarga Group for all the three limbs has to be regarded as syn-folding. With the exception of few bedding-oblique strands
of flexural-slip faults, no significant evidence of strata thinning/thickening has been observed in the Vallcarga group. This





points out that, folding has been almost entirely produced by layer parallel-shearing with bed-thickness preservation until
late stage flexural-slip faulting took place (e.g. Donath and Parker, 1964). (3) Deformation structures such as the layer-
(nearly) parallel shortening related cleavage measured in the silty levels of the Areny Group along the axial zone of the San
Maximí syncline indicate a maximum stress oriented at a low angle to bedding (Fig. 2d). (4) The fourth key observation
concerns the steeply-dipping faults, with high cutoff angles, cutting and displacing the unconformity. These faults include
steps with cut-off angles of about 30° and steps parallel to the overturned bedding surfaces. Under the assumption that late-
stage flexural slip faulting caused the arrest of shearing along the unconformity, a range of possible maximum stress
orientation during the transition from top-to-the-foreland shearing along the unconformity to the late-stage flexural-slip
faulting can be defined for the central limb, as shown in figure 7b. The angle between the maximum stress and the bedding-
parallel steps of flexural-slip faults in the Vallcarga Group is $\alpha$, the angle between the maximum stress and the flexural-slip
fault strands oblique to bedding in the Vallcarga Group is $\beta$, whereas the angle between the maximum stress and the
unconformity is $\gamma$. These three angles must be comprised between 0 and 90° to produce the observed shear pattern and for it
to be kinematically compatible. When using the average dip of the unconformity (i.e. 0°) and the dip of the Vallcarga Group
strata in the central limb (i.e. 60° overturned), a maximum stress dip (labelled $d\sigma_1$ in figure 7d) ranging from 30° to 90° is
obtained.
At this stage one may argue that the maximum stress was inclined only during the latest stage of folding, when pre-
kinematic strata were overturned, while the maximum stress was sub-horizontal during most of the folding process. Such a
scenario, in which the reorientation of the maximum stress is a discontinuous process, contrasts with the fact that, in order to
produce shearing along bedding surfaces, the maximum stress should have been south-dipping not only when strata were 60°
overturned. In fact, dip-slip shearing along upright layers also requires the maximum stress to be south-dipping, and such a
stress configuration can be extrapolated also for steeply (e.g. >75°) south-dipping strata. In agreement, it is intuitive the
conclusion that the stress rotation was not a discontinuous process, but instead it has continuously operated during folding.
As documented in Tavani et al (2011), the layer parallel shortening pattern in the the Sant Corneli-Bóixols Anticline
indicates a sub-horizontal maximum stress before folding and during the early stages of folding. As evidenced by data
presented here, the stress was in a configuration not allowing shearing (referring to faulting in the Vallcarga layers) during
almost the entire folding process. Such stress configuration was able to produce shearing along the bedding surfaces and
along the unconformity, with faulting in the Vallcarga and Areny groups being almost negligible. Apart from those structures
associated with the layer-parallel shearing, the few additional deformation structures point out for a maximum stress oriented
at low angle to bedding. During the late stage of folding instead, when the unconformity angle exceeded 120-130°, the
maximum stress was south-dipping, with an angle higher than 30°. The stress attained a state allowing shearing, as faulting
in the Vallcarga started. Contextually, bedding surfaces of the Vallcarga continued to be sheared, while shearing along the
unconformity (in the central limb) arrested.



### 6.3 Flexural-slipping and stress reorientation

The information discussed above evidences for a syn-folding maximum stress rotation/reorientation within the growing anticline, from sub-horizontal early-folding layer-parallel shortening in Tavani et al (2011) to south-dipping maximum stress in overturned strata documented here. As schematically illustrated in figure 7d, we infer that the sub-horizontal maximum stress applied to the leading syncline of the growing Sant Corneli-Bóixols Anticline (i.e. the remote applied stress has an andersonian compressive configuration; Anderson, 1951), progressively rotated as it was transmitted across folding rock volumes affected by widespread flexural-slipping. In agreement with what illustrated in the introduction, here this process of stress deflection is interpreted as associable with the flexural-slipping mechanism. In fact, as largely documented, shearing along low-friction faults produces the perturbation of the remotely applied stress field (e.g. Pollard and Segall, 1987; Soliva et al., 2010) and, in particular, reduction of the fault-parallel shear stress component causes the orientation of principal stress to rotate locally towards a fault-parallel direction. Consistently with this, the coupling between flexural-slip and maximum stress reorientation documented in other structures, has been attributed to the fact that slipping along closely spaced low friction bedding surface imposes the maximum stress to orient at low angle to the slipping bedding surface in a wide area (i.e. the flexural-slip folded area), as mentioned in the introduction (Wiltschko et al., 1985; Ohlmacher and Aydin, 1997; Tavani et al. 2012). This concept fully applies to the data presented in this work until the strata attained a strongly overturned attitude.

### 7 Conclusions

The key questions in this study were to constrain up to the range of dip values over which the flexural slip mechanism can operate.. This fundamental information allows to discriminate whether flexural-slipping is a nearly passive process with respect to the stress field evolution (i.e. the remotely applied stress is only locally deflected) or rather, it is a fully active process, discontinuous at the local scale but sufficiently dense in time and space at the scale of the fold, able to reorient the remotely applied stress field. The answer is that flexural-slip in the studied structures was an active process. It operated up to 120° of dip and caused the maximum stress to progressively reorient at low angle to bedding until strata attained an overturned attitude.

### Acknowledgments

The authors sincerely thanks the reviewers, Ryan Shackleton and Juliet Crider, and the editors, for their constructive comments and suggestions. All the structural data and the photographs presented in this work can be obtained from the corresponding author. The cross sections presented in this work have been constructed using the 3D Move software. This work is a contribution of the Institut de Recerca Geomodels and the Geodinàmica i Analisi de Conques research group (2014SGR467SGR) from the Agència de Gestió d'Ajuts Universitaris i de Recerca (AGAUR) and the Secretaria d'Universitats i Recerca del Departament d'Economia i Coneixement de la Generalitat de Catalunya. Move restoration software was used for cross-section construction.

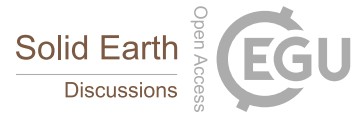

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

**Captions**
**Figure 1**: (A) Geological maps of eastern Pyrenees, with detail of the Sant Corneli-Bóixols Anticline. Geological map (B) and schematic
cross-section (C) of the study area, with cumulative contouring of poles to bedding and best-fit beta axis. (D) Panoramic view of the study
area, with insets showing the location of figures 2 to 4 and the photographed area in cross-section and map-view, respectively.
**Figure 2**: Structures exposed at the northern limb of the San Maximí Syncline. (A) Cross-sectional location of the site. (B) South-dipping
conglomerates of the Areny Group unconformably overlying the overturned north-dipping strata of the Vallcarga Group, with stereoplots
of the unconformity and bedding surfaces in the Vallcarga Group. (C) Detail of the unconformity, showing S-C-C' fabric, with
corresponding stereoplots. (D) South-dipping alternating conglomerates and siltstones of the Areny Group, with pervasive cleavage at high
angle to bedding.
**Figure 3:** Transitional area between the San Maximí Syncline and the Remolina Anticline. (A) Cross-sectional location of the site. (B)
Shallow-dipping unconformity between the Areny and the Vallcarga groups reactivated as a low-angle fault and displaced by a high-angle
fault. Details of the low- and high-dipping faults are shown, together with stereoplots of faults and bedding surfaces of the Vallcarga
Group. (C) Panoramic view and stereoplot of a near vertical fault system uplifting the Remolina Anticline, which has folded strata of the
Areny Group unconformably on top of near vertical strata of the Vallcarga Group.
**Figure 4:** Macro- and meso-structures of the Remolina Anticline. (A) Panoramic view and line-drawing of the Remolina Anticline (with
insets showing the location of figure 4B and C, and figure 5B), with stereoplots of faults and bedding measured in the Vallcarga Group. (B)
Detail of a south-dipping reverse fault having in its footwall sub-horizontal carbonates of the Areny Group on top of overturned strata of
the Vallcarga Group. Stereoplots show fault data in the Areny carbonates. (C) Details of the unconformity, with S-C-C' illustrated and
plotted.
**Figure 5:** (A) Panoramic view of the hinge zone of the Remolina anticline, visible in the Areny strata that are on top of constantly-dipping
strata of the Vallcarga Group. (B) Detail of the unconformity between Areny and Vallcarga groups at the southern limb of the anticline,
where no evidence of shear occurs. (C) Detail of slickenlines along a near vertical bedding surface of the Areny strata, providing a top to
the north shear sense for the upper bed. (D) Stereoplots of bedding surfaces and faults collected in the Areny Group strata of the southern
limb of the Remolina Anticline.
**Figure 6:** (A) Evolving angular relationships between unconformable sequences during flexural-folding in the inner limb of a syncline,
with shear senses along pre-unconformity layers and along the unconformity indicated, for different initial unconformity angles. The
position of six key-points undergoing folding is illustrated, as well as the dip of pre-unconformity layers ($D$) and of the unconformity angle
($U$), which is the angle between the unconformity and the underlying layers. (B) X and Y coordinates of the six points of figure 6a, with
length of segments, and derived amount of shear along the unconformity ($\Delta S$) and unconformity angle (U). (C) Graphical solution of
equations in figure 6b. Blue lines relate the unconformity angle ($U$) to the dip of pre growth strata ($D$) for different initial unconformity
angle ($U_0$). Red lines relates the normalised shear along the unconformity in the inner portion (i.e. $\Delta S = P_4P_5$ segment divided $H$) to $D$, for
different initial unconformity angle ($U_0$). Notice that the Y axis for red lines is on the right and that positive and negative values are





flipped. The lines indicate the cumulative shear along the unconformity, while the grey area bordered by the black line, indicate the area
where the incremental shear is negative.
**Figure 7:** (A) Scheme showing the present day geometry of the frontal limb of the Sant Corneli-Bóixols Anticline. (B) Details showing the
structural assemblages observed at the Remolina Anticline, with two alternative configurations for the maximum stress orientation. The
maximum stress forms the following clockwise angles: $d\sigma_1$ with the horizontal, $\gamma$ with the unconformity, $\alpha$ with the bedding-parallel steps
of flexural-slip faults in the Vallcarga Group, $\beta$ with the oblique to bedding strands of the flexural-slip fault in the Vallcarga Group. Red
and cyan colours indicate angles not compatible and compatible with the observed shear pattern, respectively. (C) Relationships between
$d\sigma_1$, and $\alpha$, $\beta$, and $\gamma$, with the red area indicating the orientation of the maximum stress not compatible with the shear pattern observed at
the Remolina Anticline. (D) Inferred maximum stress trajectories during the late stages of folding.





Figure 1: (A) Geological maps of eastern Pyrenees, with detail of the Sant Corneli-Bóixols Anticline. Geological map (B) and schematic cross-section (C) of the study area, with cumulative contouring of poles to bedding and best-fit beta axis. (D) Panoramic view of the study area, with insets showing the location of figures 2 to 4 and the photographed area in cross-section and map-view, respectively.





Figure 2: Structures exposed at the northern limb of the San Maximí Syncline. (A) Cross-sectional location of the site. (B) South-dipping conglomerates of the Areny Group unconformably overlying the overturned north-dipping strata of the Vallcarga Group, with stereoplots of the unconformity and bedding surfaces in the Vallcarga Group. (C) Detail of the unconformity, showing S-C-C' fabric, with corresponding stereoplots. (D) South-dipping alternating conglomerates and siltstones of the Areny Group, with pervasive cleavage at high angle to bedding.

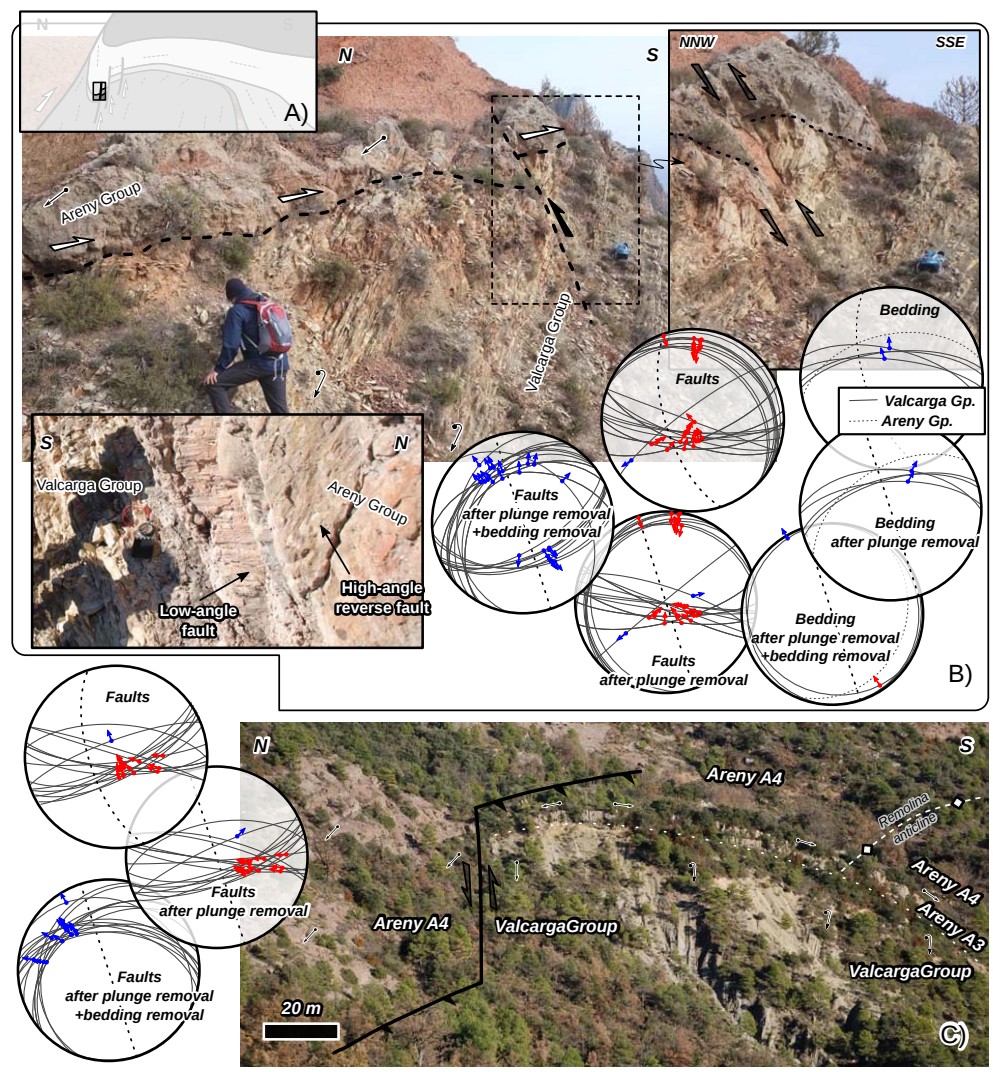

Figure 3: Transitional area between the San Maximí Syncline and the Remolina Anticline. (A) Cross-sectional location of the site. (B) Shallow-dipping unconformity between the Areny and the Vallcarga groups reactivated as a low-angle fault and displaced by a high-angle fault. Details of the low- and high-dipping faults are shown, together with stereoplots of faults and bedding surfaces of the Vallcarga Group. (C) Panoramic view and stereoplot of a near vertical fault system uplifting the Remolina Anticline, which has folded strata of the Areny Group unconformably on top of near vertical strata of the Vallcarga Group.



Figure 4: Macro- and meso-structures of the Remolina Anticline. (A) Panoramic view and line-drawing of the Remolina Anticline (with insets showing the location of figure 4B and C, and figure 5B), with stereoplots of faults and bedding measured in the Vallcarga Group. (B) Detail of a south-dipping reverse fault having in its footwall sub-horizontal carbonates of the Areny Group on top of overturned strata of the Vallcarga Group. Stereoplots show fault data in the Areny carbonates. (C) Details of the unconformity, with S-C-C' illustrated and plotted.

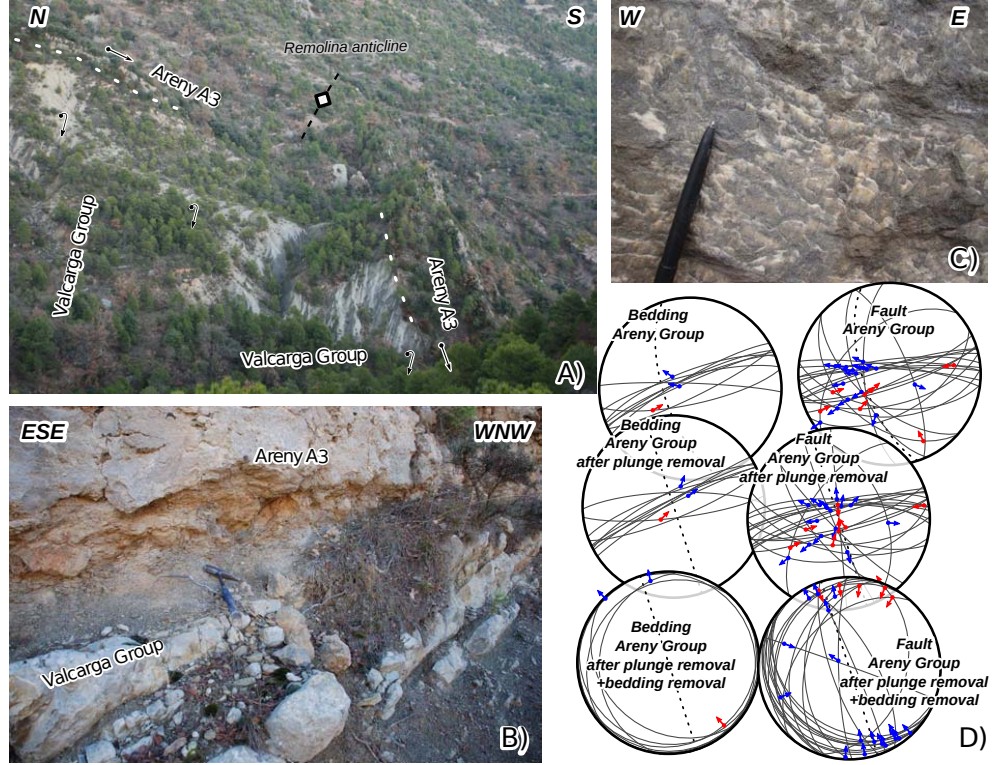

Figure 5: (A) Panoramic view of the hinge zone of the Remolina anticline, visible in the Areny strata that are on top of constantly-dipping strata of the Vallcarga Group. (B) Detail of the unconformity between Areny and Vallcarga groups at the southern limb of the anticline, where no evidence of shear occurs. (C) Detail of slickenlines along a near vertical bedding surface of the Areny strata, providing a top to the north shear sense for the upper bed. (D) Stereoplots of bedding surfaces and faults collected in the Areny Group strata of the southern limb of the Remolina Anticline.





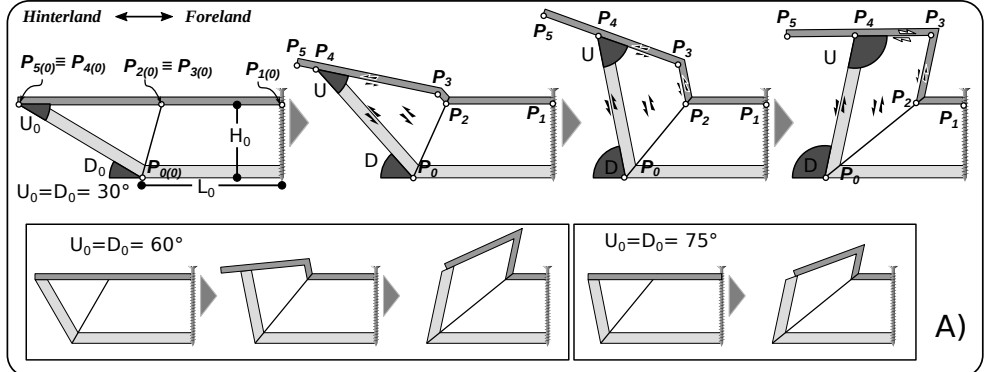

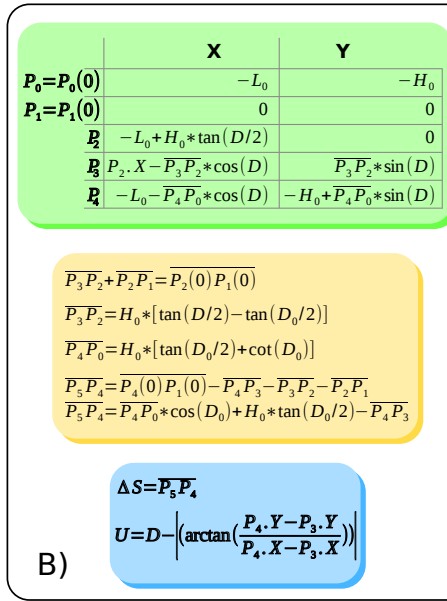

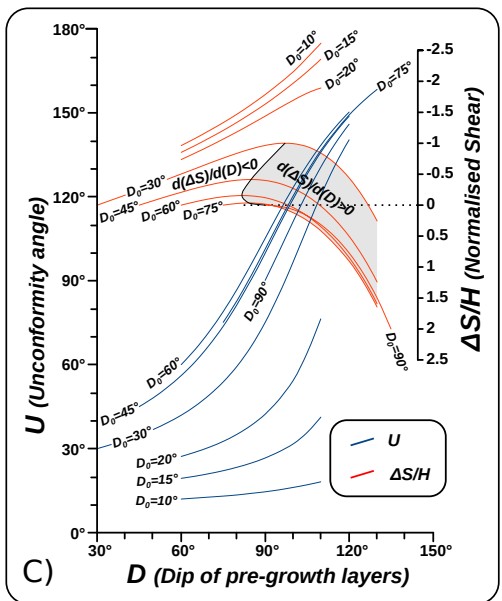

Figure 6: (A) Evolving angular relationships between unconformable sequences during flexural-folding in the inner limb of a syncline, with shear senses along pre-unconformity layers and along the unconformity indicated, for different initial unconformity angles. The position of six key-points undergoing folding is illustrated, as well as the dip of pre-unconformity layers (D) and of the unconformity angle (U), which is the angle between the unconformity and the underlying layers. (B) X and Y coordinates of the six points of figure 6a, with length of segments, and derived amount of shear along the unconformity (ΔS) and unconformity angle (U). (C) Graphical solution of equations in figure 6b. Blue lines relate the unconformity angle (U) to the dip of pre growth strata (D) for different initial unconformity angle (U0). Red lines relates the normalised shear along the unconformity in the inner portion (i.e. ΔS = P4P5 segment divided H) to D, for different initial unconformity angle (U0). Notice that the Y axis for red lines is on the right and that positive and negative values are flipped. The lines indicate the cumulative shear along the unconformity, while the grey area bordered by the black line, indicate the area where the incremental shear is negative.



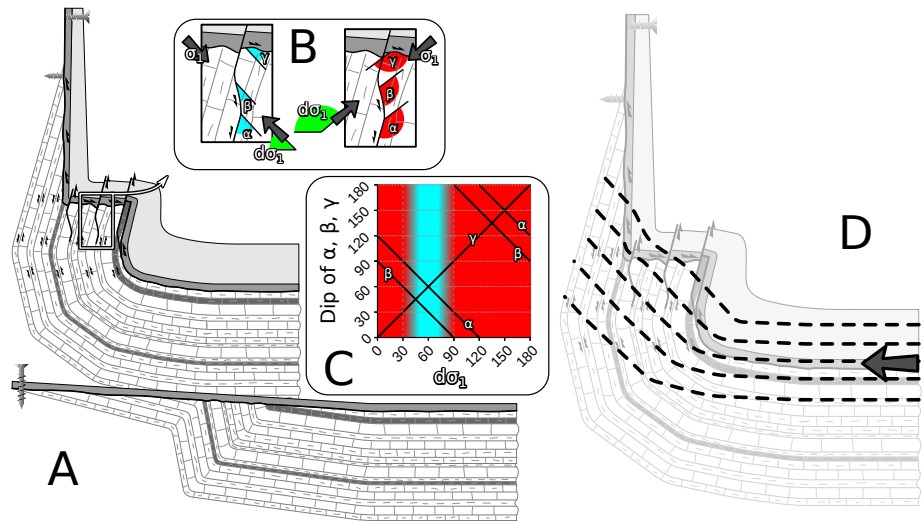

Figure 7: (A) Scheme showing the present day geometry of the frontal limb of the Sant Corneli-Bóixols Anticline. (B) Details showing the structural assemblages observed at the Remolina Anticline, with two alternative configurations for the maximum stress orientation. The maximum stress forms the following clockwise angles: $d\sigma_1$ with the horizontal, $\lambda$ with the unconformity, $\alpha$ with the bedding-parallel steps of flexural-slip faults in the Vallcarga Group, $\beta$ with the oblique to bedding strands of the flexural-slip fault in the Vallcarga Group. Red and cyan colours indicate angles not compatible and compatible with the observed shear pattern, respectively. (C) Relationships between $d\sigma_1$, and $\alpha$, $\beta$, and $\lambda$, with the red area indicating the orientation of the maximum stress not compatible with the shear pattern observed at the Remolina Anticline. (D) Inferred maximum stress trajectories during the late stages of folding.