# Peer review of "Syn-thrusting, near-surface flexural-slipping and stress deflection"

_Solid Earth, 2017_

## Short Comment (SC1) · 6 Feb 2017

Tavani et al 2017 Solid Earth

Presentation Text Line 13- S-shaped fold? 15-shallowly»gently 28-broad? 29- to model » for modelling 32 consequences on » consequences for 33 in a wide » with a wide 36 imposes » constrains 43 with the direction » so that the direction 43 maximum stress » maximum compressive stress 43 keeping at low able to layers » maintains a low angle to the bedding 43 to layers »to the bedding 44 maybe not » may not be 48 come uniquely »come 49 deciphering for which dip values a given »determining the bedding

dip when a given 51 allow to drastically reducing »serve to drastically reduce 52 As a matter of fact, » In fact, 56 few » only a few 59 macro and » macro- and 61 pre and » pre- and 63 anisotropies oblique to each other » oblique anisotropies 78 were carried » have been carried 82 transition » change 95 evidences »indications 113 at the northern limb » on the northern limb 114 The upper one » The upper thrust 121 strike » plunge direction 133 In the upper » On the upper 141 affects few » affects a few 145 in correspondence of » associated with 148 is at high angle » is at a high angle 151 lay » lie 162 for »of 166 layers » bedding 167 block is few »block is a few 173 These faults are at » The faults are at a 181 MEANING NOT CLEAR 184 reduced down »reduced 188 Faults at low angle » Faults at a low angle 191 provided » exhibited 217 and the bedding » and the dip of the bedding 221 on top of » overlying the 223 Such shear » Such a shear 224 The NNW- » However the NNW- 225 anticline tough. » anticline. 236 where » were 236 firstly provided » first given 239 six key-points » six material points 299 points out » suggests 359 evidences for» argues for

Figures Fig 2a is hardly visible. Fig 2b hook-shaped symbols are strange. What do they signify? Fig 2 more complete caption needed to explain the symbols used.  

Geological points 1) Line 214 and Fig2d - bedding and cleavage indicates that outcrop is on the S limb of an antiform. Doesn't agree with the cross section. 2) 198 Restoration is carried out by first correcting for the fold axis plunge (fig 2 and 3). This assumes that folds were non-plunging initially. Is there evidence? 3) 227-233 The idea of flexural - slip folds accommodating shortening in a direction which not perpendicular to the hinge line is interesting. Does it work? Once the fold had initiated, slip would be difficult in an oblique direction? 4) Ramsay 1967 p494 seems relevant to your paper. 5) In Figure 6a it would be helpful to know which points are "material points", i.e. attached to the rock, and which are migrating through the rock, e.g. P2??. 6) Is P3 a material point? 7) What about the stretching of the unconformity surface indicated in Fig 6a? 8) The axial surface migrates through the material, therefore expect complex strain history of the limbs concerned. Why should ax surf be located there? 9) Fig 6a, (top row, far right):

The indicated sense of shear on the unconformity does not agree with the displacement of Point P4 10) Taking the equation for cumulative shear and measuring P5-P4 gives an increase in shear from left to right. This does not agree with your statement that the sign of incremental shear changes. 11) Lines 248-254. Equation 3-92 on page 102 Ramsay 1967 explains the change of sign of the infinitesimal shear strain. From this you can see that sense of shear changes once the unconformity becomes 135° with the shear plane (the bedding plane in the flexural slip folds).

12) The equations in Fig 6a assume that the length of the Arenys rocks measured along the unconformity is conserved. However the cleavage in Fig 2d implies that there is a penetrative strain of the younger series, at least at some places.

13) 327-329 Cleavage is normally considered to be a finite strain structure. Its relationship with stress will usually be complex. Cleavage requires significant strains. The means that constant bed length assumptions are suspect, at least locally.

14) 369-373. The discussion of stress orientations is difficult especially in flexural slip folds. Probable the orientation of sigma1 rotates relative to bedding repeatedly during pulse of flexural slip.

---

## Author Comment (AC1) · 9 Feb 2017

We have uploaded the response to the reviewer's comments, the manuscript with changes marked, and the new figures (figs 1,2,3 and 6) as supplementary material

Please also note the supplement to this comment:
http://www.solid-earth-discuss.net/se-2017-2/se-2017-2-AC1-supplement.zip

---

## Referee Comment (RC2) · H. Ortner (Referee) · 21 Feb 2017

This paper presents the results of a field study of brittle faulting mostly related to flexural slip in a growth syncline and along the principal unconformity. The fact that a large parts oft he involved sediments were deposited during folding allow a good temporal resolution of formation of structures. While the study is well written and nicely documented, I have a few suggestions to improve the manuscript:

(1) In the introduction, I miss a regional cross section to illustrate the larger scale structures shown on the map in figure 1A, namely the Tremp syncline, Boixols anticline,

and Santa Fe syncline. This section would both illustrate the nature of the main ramp (line 107) and the inverted rift structures oft he Organya basin.

(2) As the authors state (lines 282-286), the geometric model (Fig. 6A) should be applied with caution. I miss a despription of reasons for applying this model, and a discussion of the discrepancies between the model and observation. I think it is important to note that the pin line in the model is in the undeformed (?) horizontal forelimb of the syncline, which would be the case for drag folds, and not in the core of the syncline, where it would be in flexural slip folds formed by layer-parallel shortening. The two cases could quickly and easily distinguished by calculating the amount of layer-parallel slip, which would be larger using a pin in theforelimb of the fold, and this could give a justification for using a specific model.

(3) On the first sight, it is very hard to understand the existence of the St. Maximi syncline and Remolina anticline, as their axial planes are parallel to bedding in the pre-folding units. It seems that these folds result from shortening perpendicular to bedding of the the pre-folding units and thus imply volume loss in these units. Alternatively, inhomogeneous flexural flow/slip in the the pre-folding units could cause this folding. Localized layer parallel slip is in the southern limb of the San Maximi syncline points to the second mode of folding. This should be clarified in the text. Are there lithologic changes in the pre-folding sediments, that could give a reason for inhomogeneous flexural slip/flow? (4) In all interpreted field photographs, arrows und j-shaped arrows parallel to bedding are shown. Indicate what these arrows mean.

Please also note the supplement to this comment:
http://www.solid-earth-discuss.net/se-2017-2/se-2017-2-RC2-supplement.pdf

**Supplement:**

[revised manuscript text omitted]

---

## Author Comment (AC2) · 24 Feb 2017

The response to the reviewer's comments, the manuscript with changes marked (those made in response to the second reviewer are highlighted in yellow), and the new figure 1 are in the supplementary material.

Please also note the supplement to this comment:
http://www.solid-earth-discuss.net/se-2017-2/se-2017-2-AC2-supplement.zip